# Reverse KL-Divergence Training of Prior Networks: Improved Uncertainty and Adversarial Robustness

**Andrey Malinin** [*]
Yandex Research
am969@yandex-team.ru

**Mark Gales**
Department of Engineering
University of Cambridge
mjfg@eng.cam.ac.uk

## Abstract

Ensemble approaches for uncertainty estimation have recently been applied to the tasks of misclassification detection, out-of-distribution input detection and adversarial attack detection. Prior Networks have been proposed as an approach to efficiently *emulate* an ensemble of models for classification by parameterising a Dirichlet prior distribution over output distributions. These models have been shown to outperform alternative ensemble approaches, such as Monte-Carlo Dropout, on the task of out-of-distribution input detection. However, scaling Prior Networks to complex datasets with many classes is difficult using the training criteria originally proposed. This paper makes two contributions. First, we show that the appropriate training criterion for Prior Networks is the *reverse* KL-divergence between Dirichlet distributions. This addresses issues in the nature of the training data target distributions, enabling prior networks to be successfully trained on classification tasks with arbitrarily many classes, as well as improving out-of-distribution detection performance. Second, taking advantage of this new training criterion, this paper investigates using Prior Networks to detect adversarial attacks and proposes a generalized form of adversarial training. It is shown that the construction of successful *adaptive* whitebox attacks, which affect the prediction and evade detection, against Prior Networks trained on CIFAR-10 and CIFAR-100 using the proposed approach requires a greater amount of computational effort than against networks defended using standard adversarial training or MC-dropout.

## 1  Introduction

Neural Networks (NNs) have become the dominant approach to addressing computer vision (CV) [1, 2, 3], natural language processing (NLP) [4, 5, 6], speech recognition (ASR) [7, 8] and bio-informatics [9, 10] tasks. One important challenge is for NNs to make reliable estimates of confidence in their predictions. Notable progress has recently been made on predictive uncertainty estimation for Deep Learning through the definition of baselines, tasks and metrics [11], and the development of practical methods for estimating uncertainty using ensemble methods, such as Monte-Carlo Dropout [12] and Deep Ensembles [13]. Uncertainty estimates derived from ensemble approaches have been successfully applied to the tasks of detecting misclassifications and out-of-distribution inputs, and have also been investigated for adversarial attack detection [14, 15]. However, ensembles can be computationally expensive and it is hard to control their behaviour. Recently, [16] proposed *Prior Networks* - a new approach to modelling uncertainty which has been shown to outperform Monte-Carlo dropout on a range of tasks. Prior Networks parameterize a Dirichlet prior over output distributions, which allows them to *emulate* an ensemble of models using a single network, whose behaviour can be *explicitly* controlled via choice of training data.

---

[*]Work done while at Cambridge University Department of Engineering

In [16], Prior Networks are trained using the *forward* KL-divergence between the model and a target Dirichlet distribution. It is, however, necessary to use auxiliary losses, such as cross-entropy, to yield competitive classification performance. Furthermore, it is also difficult to train Prior Networks using this criterion on complex datasets with many classes. In this work we show that the *forward* KL-divergence (KL) is an inappropriate optimization criterion and instead propose to train Prior Networks with the *reverse* KL-divergence (RKL) between the model and a target Dirichlet distribution. In sections 3 and 4 of this paper it is shown, both theoretically and empirically on synthetic data, that this loss yields the desired behaviours of a Prior Network and does not require auxiliary losses. In section 5 Prior Networks are successfully trained on a range of image classification tasks using the proposed criterion without loss of classification performance. It is also shown that these models yield better out-of-distribution detection performance on the CIFAR-10 and CIFAR-100 datasets than Prior Networks trained using *forward* KL-divergence.

An interesting application of uncertainty estimation is the detection of *adversarial attacks*, which are small perturbations to the input that are almost imperceptible to humans, yet which drastically affect the predictions of the neural network [17]. Adversarial attacks are a serious security concern, as there exists a plethora of adversarial attacks which are quite easy to construct [18, 19, 20, 21, 22, 23, 24, 25]. At the same time, while it is possible to improve the robustness of a network to adversarial attacks using adversarial training [17] and adversarial distillation [26], it is still possible to craft successful adversarial attacks against these networks [21]. Instead of considering *robustness* to adversarial attacks, [14] investigates *detection* of adversarial attacks and shows that adversarial attacks can be detectable using a range of approaches. While, *adaptive* attacks can be crafted to successfully attack the proposed detection schemes, [14] singles out detection of adversarial attacks using uncertainty measures derived from Monte-Carlo dropout as being more challenging to successfully overcome using adaptive attacks. Thus, in this work we investigate the detection of adversarial attacks using Prior Networks, which have previously outperformed Monte-Carlo dropout on other tasks.

Using the greater degree of control over the behaviour of Prior Networks which the *reverse* KL-divergence loss affords, Prior Networks are trained to predict the correct class on adversarial inputs, but yield a higher measure of uncertainty than on natural inputs. Effectively, this is a direct generalization of adversarial training [17] which improves *both* the robustness of the model to adversarial attacks *and* also allows them to be detected. As Prior Networks yield measures of uncertainty derived from distributions over output distributions, rather than simple confidences, adversarial attacks need to satisfy far more constraints in order to both successfully attack the Prior Network and evade detection. Results in section 6 show that on the CIFAR-10 and CIFAR-100 datasets it is more computationally challenging to construct *adaptive* adversarial attacks against Prior Networks than against standard DNNs, adversarially trained DNNs and Monte-Carlo dropout defended networks.

Thus, the two main contributions of this paper are the following. Firstly, a new *reverse* KL-divergence training criterion which yields the desired behaviour of Prior Networks and allows them to be trained on more complex datasets. Secondly, a generalized form of adversarial training, enabled using the proposed training criterion, which makes successful *adaptive* whitebox attacks, which aim to both attack the network and evade detection, far more computationally expensive to construct for Prior Networks than for models defended using standard adversarial training or Monte-Carlo dropout.

## 2 Prior Networks

An ensemble of models can be interpreted as a set of output distributions drawn from an *implicit* conditional distribution over output distributions. A Prior Network $p(\boldsymbol{\pi}|\boldsymbol{x}^*; \hat{\boldsymbol{\theta}})$ [2], is a neural network which *explicitly* parametrizes a prior distribution over output distributions. This effectively allows a Prior Network to *emulate* an ensemble and yield the same measures of uncertainty [27, 28], but in closed form and without sampling.

$$p(\boldsymbol{\pi}|\boldsymbol{x}^*; \hat{\boldsymbol{\theta}}) = p(\boldsymbol{\pi}|\hat{\boldsymbol{\alpha}}), \quad \hat{\boldsymbol{\alpha}} = \boldsymbol{f}(\boldsymbol{x}^*; \hat{\boldsymbol{\theta}}) \quad (1)$$

A Prior Network typically parameterizes the Dirichlet distribution[3] (eq. 2), which is the conjugate prior to the categorical, due to its tractable analytic properties. The Dirichlet distribution is defined

as:

$$\mathrm{p}(\boldsymbol{\pi}; \boldsymbol{\alpha}) = \frac{\Gamma(\alpha_0)}{\prod_{c=1}^{K} \Gamma(\alpha_c)} \prod_{c=1}^{K} \pi_c^{\alpha_c - 1}, \quad \alpha_c > 0, \ \alpha_0 = \sum_{c=1}^{K} \alpha_c \qquad (2)$$

where $\Gamma(\cdot)$ is the gamma function. The Dirichlet distribution is parameterized by its concentration parameters $\boldsymbol{\alpha}$, where $\alpha_0$, the sum of all $\alpha_c$, is called the *precision* of the Dirichlet distribution. Higher values of $\alpha_0$ lead to sharper, more confident distributions. The predictive distribution of a Prior Network is given by the expected categorical distribution under the conditional Dirichlet prior:

$$\mathrm{P}(y = \omega_c | \boldsymbol{x}^*; \hat{\boldsymbol{\theta}}) = \mathbb{E}_{\mathrm{p}(\boldsymbol{\pi}|\boldsymbol{x}^*; \hat{\boldsymbol{\theta}})}[\mathrm{P}(y = \omega_c | \boldsymbol{\pi})] = \hat{\pi}_c = \frac{\hat{\alpha}_c}{\sum_{k=1}^{K} \alpha_k} = \frac{e^{\hat{z}_c}}{\sum_{k=1}^{K} \hat{e}^{\hat{z}_k}} \qquad (3)$$

where $\hat{z}_c$ are the logits predicted by the model. The desired behaviors of a Prior Network, as described in [16], can be visualized on a simplex in figure 1. Here, figure 1:a describes confident behavior (low-entropy prior focused on low-entropy output distributions), figure 1:b describes uncertainty due to severe class overlap (*data uncertainty*) and figure 1:c describes the behaviour for an out-of-distribution input (*knowledge uncertainty*).

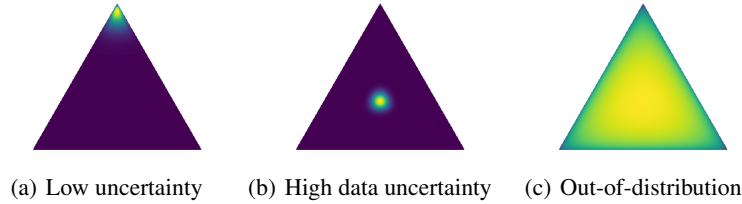

    (a) Low uncertainty      (b) High data uncertainty      (c) Out-of-distribution

Figure 1: Desired Behaviors of a Dirichlet distribution over categorical distributions.

Given a Prior Network which yields the desired behaviours, it is possible to derive measures of uncertainty in the prediction by considering the mutual information between $y$ and $\boldsymbol{\pi}$:

$$\underbrace{\mathcal{MI}[y, \boldsymbol{\pi}|\boldsymbol{x}^*; \hat{\boldsymbol{\theta}}]}_{Knowledge \ Uncertainty} = \underbrace{\mathcal{H}\big[\mathbb{E}_{\mathrm{p}(\boldsymbol{\pi}|\boldsymbol{x}^*; \hat{\boldsymbol{\theta}})}[\mathrm{P}(y|\boldsymbol{\pi})]\big]}_{Total \ Uncertainty} - \underbrace{\mathbb{E}_{\mathrm{p}(\boldsymbol{\pi}|\boldsymbol{x}^*; \hat{\boldsymbol{\theta}})}\big[\mathcal{H}[\mathrm{P}(y|\boldsymbol{\pi})]\big]}_{Expected \ Data \ Uncertainty} \qquad (4)$$

The given expression allows *total uncertainty*, given by the entropy of the predictive distribution, to be decomposed into *data uncertainty* and *knowledge uncertainty*. *Data uncertainty* arises due to class-overlap in the data, which is the equivalent of noise for classification problems. *Knowledge Uncertainty*, also know as *epistemic uncertainty* [12] or *distributional uncertainty* [16], arises due to the model's lack of understanding or *knowledge* about the input. In other word, *knowledge uncertainty* arises due to a mismatch between the training and test data.

## 3 Forward and Reverse KL-Divergence Losses

As Prior Networks parameterize the Dirichlet distribution, ideally we would like to have a dataset $\mathcal{D}_{trn} = \{\boldsymbol{x}^{(i)}, \boldsymbol{\beta}^{(i)}\}_{i=1}^{N}$, where $\boldsymbol{\beta}^{(i)}$ are the parameters of a target Dirichlet distribution $\mathrm{p}(\boldsymbol{\pi}|\boldsymbol{\beta})$. In this scenario, we could simply minimize the (forward) KL-divergence between the model and the target for every training sample $\boldsymbol{x}^{(i)}$. Alternatively, if we had a set of *samples* of categorical distributions from the target Dirichlet distribution for *every input*, then we could maximize the likelihood their under the predicted Dirichlet [29], which, in expectation, is equivalent to minimizing the KL-divergence. In practice, however, we only have access to the target class label $y^{(i)} \in \{\omega_1, \cdots, \omega_K\}$ for every input $\boldsymbol{x}^{(i)}$. When training standard DNNs with cross-entropy loss this isn't

a problem, as the correct target distribution $\hat{P}_{tr}(y|\boldsymbol{x})$ is *induced in expectation*, as shown below:

$$
\begin{aligned}
\mathcal{L}(\boldsymbol{\theta}, \mathcal{D}_{trn}) &= \mathbb{E}_{\hat{p}_{tr}(\boldsymbol{x})}\Big[-\sum_{c=1}^{K}\mathbb{E}_{\hat{P}_{tr}(y|\boldsymbol{x})}[\mathcal{I}(y=\omega_c)]\ln P(\hat{y}=\omega_c|\boldsymbol{x};\boldsymbol{\theta})\Big] \\
&= \mathbb{E}_{\hat{p}_{tr}(\boldsymbol{x})}\Big[-\sum_{c=1}^{K}\hat{P}_{tr}(y=\omega_c|\boldsymbol{x})\ln P(\hat{y}=\omega_c|\boldsymbol{x};\boldsymbol{\theta})\Big] \\
&= \mathbb{E}_{\hat{p}_{tr}(\boldsymbol{x})}\Big[\mathrm{KL}\big[\hat{P}_{tr}(y|\boldsymbol{x})||P(y|\boldsymbol{x};\boldsymbol{\theta})\big]\Big]+const \\
&= \mathbb{E}_{\hat{p}_{tr}(\boldsymbol{x})}\Big[\mathrm{KL}\big[\boldsymbol{\pi}_{tr}||\hat{\boldsymbol{\pi}}\big]\Big]+const
\end{aligned}
\tag{5}
$$

Unfortunately, training models which are a (higher-order) *distribution over predictive distributions* based on samples from the (lower-order) predictive distribution is more challenging. The solution to this problem proposed in the original work on Prior Networks [16] was to minimize the *forward* KL-divergence between the model and a target Dirichlet distribution $p(\boldsymbol{\pi}|\boldsymbol{\beta}^{(c)})$:

$$
\mathcal{L}^{KL}(y, \boldsymbol{x}, \boldsymbol{\theta}; \beta) = \sum_{c=1}^{K}\mathcal{I}(y=\omega_c)\cdot\mathrm{KL}[p(\boldsymbol{\pi}|\boldsymbol{\beta}^{(c)})||p(\boldsymbol{\pi}|\boldsymbol{x};\boldsymbol{\theta})]
\tag{6}
$$

The target concentration parameters $\boldsymbol{\beta}^{(c)}$ depend on the class $c$ and are set manually as follows:

$$
\beta_k^{(c)} = \begin{cases} \beta+1 & if\ c=k \\ 1 & if\ c\neq k \end{cases}
\tag{7}
$$

where $\beta$ is a *hyper-parameter* which is set by hand, rather than learned from the data. This criterion is jointly optimized on in-domain and out-of-domain data $\mathcal{D}_{trn}$ and $\mathcal{D}_{out}$ as follows:

$$
\mathcal{L}(\boldsymbol{\theta}, \mathcal{D}; \beta_{in}, \beta_{out}, \gamma) = \mathcal{L}_{in}^{KL}(\boldsymbol{\theta}, \mathcal{D}_{trn}; \beta_{in}) + \gamma\cdot\mathcal{L}_{out}^{KL}(\boldsymbol{\theta}, \mathcal{D}_{out}; \beta_{out})
\tag{8}
$$

where $\gamma$ is the out-of-distribution loss weight. In-domain $\beta_{in}$ should take on a large value, for example $1e2$, so that the concentration is high only in the corner corresponding to the target class, and low elsewhere. Note, the concentration parameters have to be strictly positive, so it is not possible to set the rest of the concentration parameters to 0. Instead, they are set to one, which also provides a small degree smoothing. Out-of-domain $\beta_{out} = 0$, which results in a flat Dirichlet distribution.

However, there is a significant issue with this criterion. Consider taking the expectation of equation 6 with respect to the empirical distribution $\hat{p}_{tr}(\boldsymbol{x}, y) = \{\boldsymbol{x}^{(i)}, y^{(i)}\}_{i=1}^{N} = \mathcal{D}_{trn}$:

$$
\begin{aligned}
\mathcal{L}^{KL}(\boldsymbol{\theta}, \mathcal{D}_{trn}; \beta) &= \mathbb{E}_{\hat{p}_{tr}(\boldsymbol{x},y)}\Big[\sum_{c=1}^{K}\mathcal{I}(y=\omega_c)\cdot\mathrm{KL}[p(\boldsymbol{\pi}|\boldsymbol{\beta}^{(c)})||p(\boldsymbol{\pi}|\boldsymbol{x};\boldsymbol{\theta})]\Big] \\
&= \mathbb{E}_{\hat{p}_{tr}(\boldsymbol{x})}\Big[\mathrm{KL}\big[\sum_{c=1}^{K}\hat{P}_{tr}(y=\omega_c|\boldsymbol{x})p(\boldsymbol{\pi}|\boldsymbol{\beta}^{(c)})||p(\boldsymbol{\pi}|\boldsymbol{x};\boldsymbol{\theta})\big]\Big]+const
\end{aligned}
\tag{9}
$$

*In expectation* this loss induces a target distribution which is a *mixture of Dirichlet distributions* that has a mode in each corner of the simplex, as shown in figure 2a. When the level of *data uncertainty* is low, this is not a problem, as there will only be a single significant mode. However, the target distribution will be multi-modal when there is a significant amount of *data uncertainty*. As the *forward* KL-divergence is *zero-avoiding*, it will drive the model to spread itself over each mode, effectively 'inverting' the Dirichlet distribution and forcing the precision $\hat{\alpha}_0$ to a low value, as depicted in figure 2b. This is an undesirable behaviour and can compromise predictive performance. Rather, as previously stated, in regions of significant *data uncertainty* the model should yield a distribution with a single high-precision mode at the center of the simplex, as shown in figure 1b.

The main issue with the *forward* KL-divergence loss is that it induces an *arithmetic mixture* of target distributions $p(\boldsymbol{\pi}|\boldsymbol{\beta}^{(c)})$ in expectation. This can be avoided by instead considering the *reverse* KL-divergence between the target distribution $p(\boldsymbol{\pi}|\boldsymbol{\beta}^{(c)})$ and the model:

$$
\mathcal{L}^{RKL}(y, \boldsymbol{x}, \boldsymbol{\theta}; \beta) = \sum_{c=1}^{K}\mathcal{I}(y=\omega_c)\cdot\mathrm{KL}[p(\boldsymbol{\pi}|\boldsymbol{x};\boldsymbol{\theta})||p(\boldsymbol{\pi}|\boldsymbol{\beta}^{(c)})]
\tag{10}
$$

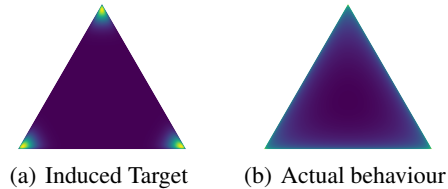

(a) Induced Target       (b) Actual behaviour

Figure 2: Induced target and predicted Dirichlet distribution when trained with equation 6

The expectation of this criterion with respect to the empirical distribution induces a *geometric mixture* of target Dirichlet distributions:

$$
\begin{aligned}
\mathcal{L}^{RKL}(\boldsymbol{\theta}, \mathcal{D}_{trn}; \beta) &= \mathbb{E}_{\hat{\mathrm{p}}_{\mathrm{tr}}(\boldsymbol{x})}\Big[ \sum_{c=1}^{K} \hat{\mathrm{P}}_{\mathrm{tr}}(y = \omega_c|\boldsymbol{x}) \mathrm{KL}\big[\mathrm{p}(\boldsymbol{\pi}|\boldsymbol{x};\boldsymbol{\theta})||\mathrm{p}(\boldsymbol{\pi}|\boldsymbol{\beta}^{(c)})\big]\Big] \\
&= \mathbb{E}_{\hat{\mathrm{p}}_{\mathrm{tr}}(\boldsymbol{x})}\Big[ \mathbb{E}_{\mathrm{p}(\boldsymbol{\pi}|\boldsymbol{x};\boldsymbol{\theta})}\big[ \ln \mathrm{p}(\boldsymbol{\pi}|\boldsymbol{x};\boldsymbol{\theta}) - \ln \prod_{c=1}^{K} \mathrm{p}(\boldsymbol{\pi}|\boldsymbol{\beta}^{(c)})^{\hat{\mathrm{P}}_{\mathrm{tr}}(y=\omega_c|\boldsymbol{x})}\big]\Big] \\
&= \mathbb{E}_{\hat{\mathrm{p}}_{\mathrm{tr}}(\boldsymbol{x})}\Big[ \mathrm{KL}\big[\mathrm{p}(\boldsymbol{\pi}|\boldsymbol{x};\boldsymbol{\theta})||\mathrm{p}(\boldsymbol{\pi}|\bar{\boldsymbol{\beta}})\big]\Big] + \mathrm{const} \\
\bar{\boldsymbol{\beta}} &= \sum_{c=1}^{K} \hat{\mathrm{P}}_{\mathrm{tr}}(y = \omega_c|\boldsymbol{x}) \cdot \boldsymbol{\beta}^{(c)}
\end{aligned}
\tag{11}
$$

A geometric mixture of Dirichlet distributions results in a standard Dirichlet distribution whose concentration parameters $\bar{\boldsymbol{\beta}}$ are an arithmetic mixture of the target concentration parameters for each class. Thus, this loss induces a target distribution which is *always* a standard uni-modal Dirichlet with a mode at the point on the simplex which reflects the correct level of *data uncertainty* (figure 1a-b). Furthermore, as a consequence using of this loss in equation 8 instead of the *forward* KL-divergence, the concentration parameters are appropriately interpolated on the boundary of the in-domain and out-of-distribution regions, where the degree of interpolation depends on the OOD loss weight $\gamma$. Further analysis of the properties of the *reverse* KL-divergence loss is provided in appendix A.

Finally, it is important to emphasize that this discussion is about *what target distribution is induced in expectation* when training models which parameterize a *distribution over output distributions* using samples from *the output distribution*. It is necessary to stress that if either the parameters of, or samples from, the correct target distribution over output distributions are available, for every input, then *forward KL-divergence* **is** a sensible training criterion.

## 4 Experiments on Synthetic Data

The previous section investigated the theoretical properties of forward and reverse KL-divergence training criteria for Prior Networks. In this section these criteria are assessed empirically by using them to train Prior Networks on the artificial high-uncertainty 3-class dataset[4] introduced in [16]. In these experiments, the out-of-distribution training data $\mathcal{D}_{out}$ was sampled such that it forms a thin shell around the training data. The target concentration parameters $\boldsymbol{\beta}^{(c)}$ were constructed as described in equation 7, with $\beta_{in} = 1e2$ and $\beta_{out} = 0$. The in-domain loss and out-of-distribution losses were equally weighted ($\gamma = 1$).

Figure 3 depicts the *total uncertainty*, *expected data uncertainty* and mutual information, which is a measure of *knowledge uncertainty*, derived using equation 4 from Prior Networks trained using both criteria. By comparing figures 3a and 3d it is clear that a Prior Network trained using *forward* KL-divergence over-estimates *total uncertainty* in domain, as the *total uncertainty* is *equally* high along the decision boundaries, in the region of class overlap and out-of-domain. The Prior Network trained using the *reverse* KL-divergence, on the other hand, yields an estimate of *total uncertainty* which better reflects the structure of the dataset. Figure 3b shows that the *expected data uncertainty* is altogether incorrectly estimated by the Prior Network trained via *forward* KL-divergence, as

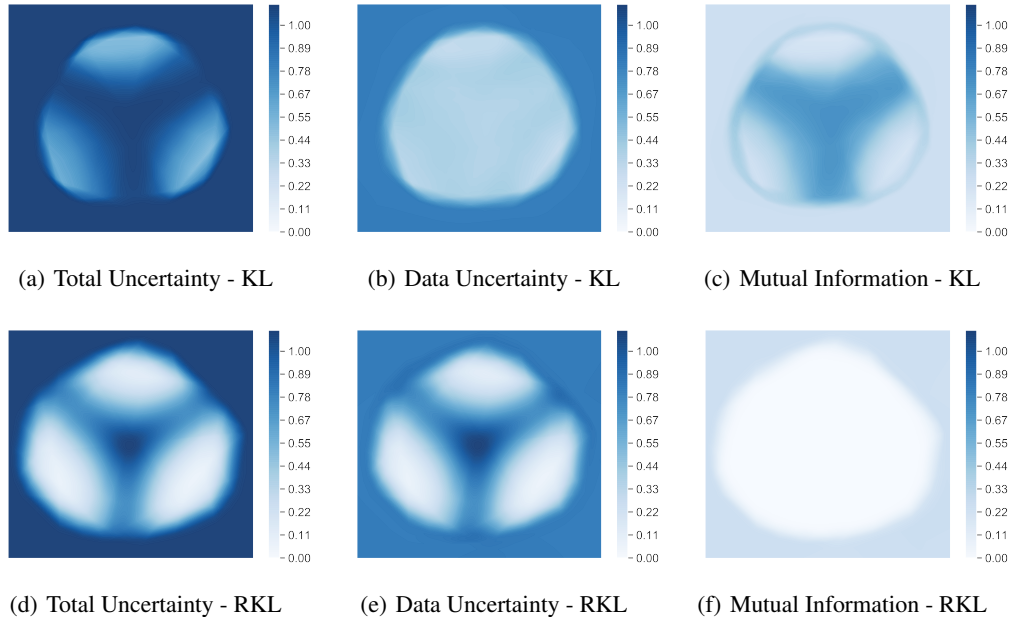

| (a) Total Uncertainty - KL | (b) Data Uncertainty - KL | (c) Mutual Information - KL |
|---|---|---|
| (d) Total Uncertainty - RKL | (e) Data Uncertainty - RKL | (f) Mutual Information - RKL |

Figure 3: Comparison of measures of uncertainty derived from Prior Networks trained with *forward* and *reverse* KL-divergence. Measures of uncertainty are derived via equation 4.

it is uniform over the entire in-domain region. As a result, the mutual information is higher in-domain along the decision boundaries than out-of-domain. In contrast, figures 3c and 3f show that the measures of uncertainty provided by a Prior Network trained using the *reverse* KL-divergence decompose correctly - *data uncertainty* is highest in regions of class overlap while mutual information is low in-domain and high out-of-domain. Thus, these experiments support the analysis in the previous section, and illustrate how the *reverse* KL-divergence is the more suitable optimization criterion.

## 5 Image Classification Experiments

Having evaluated the *forward* and *reverse* KL-divergence losses on a synthetic dataset in the previous section, we now evaluate these losses on a range of image classification datasets. The training configurations are described in appendix C. Table 1 presents the classification error rates of standard DNNs, an ensemble of 5 DNNs [13], and Prior Networks trained using both the *forward* and *reverse* KL-divergence losses. From table 1 it is clear that Prior Networks trained using *forward* KL-divergence (PN-KL) achieve increasingly worse classification performance as the datasets become more complex and have a larger number of classes. At the same time, Prior Networks trained using the *reverse* KL-divergence loss (PN-RKL) have similar error rates as ensembles and standard DNNs. Note that in these experiments no auxiliary losses were used.[5]

Table 1: Mean classification performance (% Error) $\pm 2\sigma$ across 5 random initializations.

| Dataset | DNN | PN-KL | PN-RKL | ENSM |
|---|---|---|---|---|
| MNIST | **0.5** $\pm 0.1$ | 0.6 $\pm 0.1$ | **0.5** $\pm 0.1$ | **0.5** $\pm$ NA |
| SVHN | 4.3 $\pm 0.3$ | 5.7 $\pm 0.2$ | 4.2 $\pm 0.2$ | **3.3** $\pm$ NA |
| CIFAR-10 | 8.0 $\pm 0.4$ | 14.7 $\pm 0.4$ | 7.5 $\pm 0.3$ | **6.6** $\pm$ NA |
| CIFAR-100 | 30.4 $\pm 0.6$ | - | 28.1 $\pm 0.2$ | **26.9** $\pm$ NA |
| TinyImageNet | 41.7 $\pm 0.4$ | - | 40.3 $\pm 0.4$ | **36.9** $\pm$ NA |

Table 2 presents the out-of-distribution detection performance of Prior Networks trained on CIFAR-10 and CIFAR-100 [30] using the *forward* and *reverse* KL-divergences. Prior Networks trained on CIFAR-10 use CIFAR-100 are OOD training data, while Prior Networks trained on CIFAR-100 use TinyImageNet [31] as OOD training data. Performance is assessed using area under an ROC curve (AUROC) in the same fashion as in [16, 11]. The results on CIFAR-10 show that PN-RKL consistently yields better performance than PN-KL and the ensemble on all OOD test datasets (SVHN, LSUN and TinyImagenet). The results using model trained on CIFAR-100 show that Prior Networks are capable of out-performing the ensembles when evaluated against the LSUN and SVHN datasets. However, Prior Networks have difficulty distinguishing between the CIFAR-10 and CIFAR-100 test sets. However, this represents a limitation of the both the classification model and the OOD training data, rather than the training criterion. Improving classification performance of Prior Networks on CIFAR-100, which improves understanding of what is 'in-domain', and using a more appropriate OOD training dataset, which provides a better contrast, is likely improve OOD detection performance.

Table 2: Out-of-domain detection results (mean % AUROC $\pm 2\sigma$ across 5 rand. inits) using mutual information (eqn. 4) derived from models trained on CIFAR-10 and CIFAR-100.

| Model | CIFAR-10 | | | CIFAR-100 | | |
|---|---|---|---|---|---|---|
| | SVHN | LSUN | TinyImageNet | SVHN | LSUN | CIFAR-10 |
| ENSM | 89.5 ± NA | 93.2 ± NA | 90.3 ± NA | 78.9 ± NA | 85.6 ± NA | **76.5** ± NA |
| PN-KL | 97.8 ±1.1. | 91.6 ±1.7 | 92.4 ±0.9 | - | - | - |
| PN-RKL | **98.2** ±1.1 | **95.7** ±0.9 | **95.7** ±0.7 | **84.8** ±0.8 | **100.0** ±0.0 | 57.8 ±0.4 |

## 6   Adversarial Attack Detection

The previous section has discussed the use of the reverse KL-divergence training criterion for training Prior Networks. Here, we show that the proposed loss also offers a generalization of adversarial training [17, 25] which allows Prior Networks to be *both* more robust to adversarial attacks *and* detect them as OOD samples. The use of measures of uncertainty for adversarial attack detection was previously studied in [14], where it was shown that Monte-Carlo dropout ensembles yield measures of uncertainty which are more challenging to attack than other considered methods. In a similar fashion to Monte-Carlo dropout, Prior Networks yield rich measures of uncertainty derived from distributions over distributions. For Prior Networks this means that for an adversarial attack to both affect the prediction and evade detection, it must satisfy several criteria. Firstly, the adversarial input must be located in a region of input space classified as the desired class. Secondly, the adversarial input must be in a region of input space where both the *relative* and *absolute* magnitudes of the model's logits $\hat{z}$, and therefore all the measures of uncertainty derivable from the predicted distribution over distribution, are the same as for the natural input, making it challenging to distinguish between the natural and adversarial input. Clearly, this places more constraints on the space of solutions for successful adversarial attacks than detection based on the confidence of the prediction, which places a constraint only on the *relative* value of just a single logit.

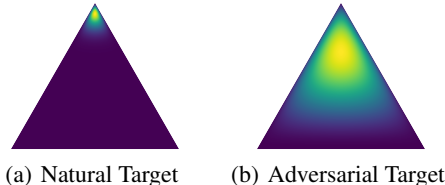

(a) Natural Target         (b) Adversarial Target

Figure 4: Target Dirichlet distributions for natural and adversarial inputs.

Using the greater degree of control over the behaviour of Prior Networks which the *reverse* KL-divergence loss affords, Prior Networks can be *explicitly* trained to yield high uncertainty for example adversarial attacks, further constraining the space of successful solutions. Here, adversarially perturbed inputs are used as the out-of-distribution training data for which the Prior Network is

trained to *both* yield the correct prediction *and* high measures of uncertainty. Thus, the Prior Network is jointly trained to yield either a *sharp* or *wide* Dirichlet distribution at the appropriate corner of the simplex for natural or adversarial data, respectively, as described in figure 4.

$$\mathcal{L}(\boldsymbol{\theta}, \mathcal{D}; \beta_{in}, \beta_{adv}, \gamma) = \mathcal{L}_{in}^{RKL}(\boldsymbol{\theta}, \mathcal{D}_{trn}; \beta_{in}) + \gamma \cdot \mathcal{L}_{adv}^{RKL}(\boldsymbol{\theta}, \mathcal{D}_{adv}; \beta_{adv}) \tag{12}$$

The target concentration parameters are set using equation 7, where $\beta_{in} = 1e2$ for natural and $\beta_{adv} = 1$ for adversarial data, for example. This approach can be seen as a generalization of *adversarial training* [17, 25]. The difference is that here we are training the model to yield a particular behaviour of an entire *distribution over output distributions*, rather than simply making sure that the decision boundaries are correct in regions of input space which correspond to adversarial attacks. Furthermore, it is important to highlight that this generalized form of adversarial training is a *drop-in replacement* for standard adversarial training which only requires changing the loss function.

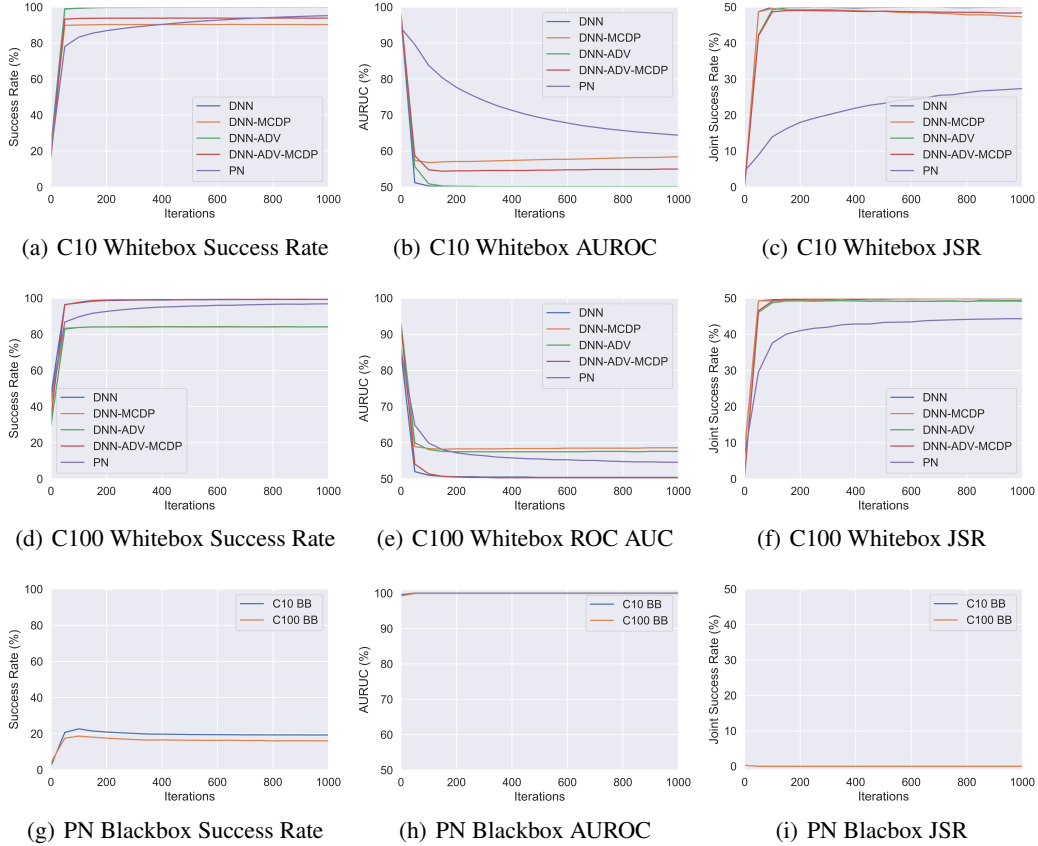

Figure 5: Adaptive Attack detection performance in terms of mean Success Rate, % AUROC and Joint Success Rate (JSR) across 5 random inits. $L_\infty$ bound on adversarial perturbation is 30 pixels.

As discussed in [14, 32], approaches to detecting adversarial attacks need to be evaluated against the strongest possible attacks - *adaptive* whitebox attacks which have full knowledge of the detection scheme and actively seek to bypass it. Here, targeted iterative PGD-MIM [20, 25] attacks are used for evaluation and simple targeted FGSM [17] are used during training. The goal is to switch the prediction to a target class but leave measures of uncertainty derived from the model unchanged.

Two forms of criteria, expressed in equation 13, are used to generate the adversarial sample, $\tilde{\boldsymbol{x}}$. For both criteria the target for the attacks is set to the second most likely class, as this should yield the least 'unnatural' perturbation of the outputs. The first approach involves permuting the model's predictive distribution over class labels $\hat{\boldsymbol{\pi}}$ and minimizing the *forward* KL-divergence between $\hat{\boldsymbol{\pi}}$ and the target permuted distribution $\boldsymbol{\pi}_{adv}$. This ensures that the target class is predicted, but places constraints only the *relative* values of the logits, and therefore only on measures of uncertainty derived from the predictive posterior. The second approach involves permuting the concentration parameters $\hat{\boldsymbol{\alpha}}$ and

minimizing the *forward* KL divergence to the permuted target Dirichlet distribution $\mathrm{p_{adv}}(\boldsymbol{\pi})$. This places constraints on both the *relative* and *absolute* values of the logits, and therefore on measures of uncertainty derived from the entire distribution over distributions.

$$\mathcal{L}_{PMF}^{KL}\big(\mathrm{P}(y|\tilde{\boldsymbol{x}};\hat{\boldsymbol{\theta}}),t\big) = \mathrm{KL}[\boldsymbol{\pi}_{adv}||\hat{\boldsymbol{\pi}}], \ \mathcal{L}_{DIR}^{KL}\big(\mathrm{p}(\boldsymbol{\pi}|\tilde{\boldsymbol{x}};\hat{\boldsymbol{\theta}}),t\big) = \mathrm{KL}[\mathrm{p_{adv}}(\boldsymbol{\pi})||\mathrm{p}(\boldsymbol{\pi}|\tilde{\boldsymbol{x}};\hat{\boldsymbol{\theta}})] \qquad (13)$$

Though $\mathcal{L}_{DIR}^{KL}$ has more explicit constraints, it was found to be more challenging to optimize and yield less aggressive attacks than $\mathcal{L}_{PMF}^{KL}$.[6] Thus, only attacks generated via $\mathcal{L}_{PMF}^{KL}$ are considered.

In the following set of experiments Prior Networks are trained on either the CIFAR-10 or CIFAR-100 datasets [30] using the procedure discussed above and detailed in appendix C. The baseline models are an undefended DNN and a DNN trained using standard adversarial training (DNN-ADV). For these models uncertainty is estimated via the entropy of the predictive posterior. Additionally, estimates of mutual information (*knowledge uncertainty*) are derived via a Monte-Carlo dropout ensemble generated from each of these models. Similarly, Prior Networks also use the mutual information (eqn. 4) for adversarial attack detection. Performance is assessed via the Success Rate, AUROC and Joint Success Rate (JSR). For the ROC curves considered here the true positive rate is computed using natural examples, while the false-positive rate is computed using only *successful* adversarial attacks[7]. The JSR, described in greater detail in appendix D, is the equal error rate where false positive rate equals false negative rate, and allows *joint* assessment of adversarial robustness and detection.

The results presented in figure 5 show that on both the CIFAR-10 and CIFAR-100 datasets whitebox attacks successfully change the prediction of DNN and DNN-ADV models to the second most likely class and evade detection (AUROC goes to 50). Monte-Carlo dropout ensembles are marginally harder to adversarially overcome, due to the random noise. At the same time, it takes far more iterations of gradient descent to successfully attack Prior Networks such that they fail to detect the attack. On CIFAR-10 the Joint Success Rate is only 0.25 at 1000 iterations, while the JSR for the other models is 0.5 (the maximum). Results on the more challenging CIFAR-100 dataset show that adversarially trained Prior Networks yield a more modest increase in robustness over baseline approaches, but it still takes significantly more computational effort to attack the model. Thus, these results support the assertion that adversarially trained Prior Networks constrain the solution space for adaptive adversarial attack, making them computationally more difficult to successfully construct. At the same time, *blackbox attacks*, computed on identical networks trained on the same data from a different random initialization, fail entirely against Prior Networks trained on CIFAR-10 and CIFAR-100. This shows that the adaptive attacks considered here are non-transferable.

## 7 Conclusion

Prior Networks have been shown to be an interesting approach to deriving rich and interpretable measures of uncertainty from neural networks. This work consists of two contributions which aim to improve these models. Firstly, a new training criterion for Prior Networks, the reverse KL-divergence between Dirichlet distributions, is proposed. It is shown, both theoretically and empirically, that this criterion yields the desired set of behaviours of a Prior Network and allows these models to be trained on more complex datasets with arbitrary numbers of classes. Furthermore, it is shown that this loss improves out-of-distribution detection performance on the CIFAR-10 and CIFAR-100 datasets relative to the *forward* KL-divergence loss used in [16]. However, it is necessary to investigate proper choice of out-of-distribution training data, as an inappropriate choice can limit OOD detection performance on complex datasets. Secondly, this improved training criterion enables Prior Networks to be applied to the task of detecting whitebox adaptive adversarial attacks. Specifically, adversarial training of Prior Networks can be seen as both a generalization of, and a drop in replacement for, standard adversarial training which improves robustness to adversarial attacks and the ability to detect them by placing more constraints on the space of solutions to the optimization problem which yields adversarial attacks. It is shown that it is significantly more computationally challenging to construct successfully adaptive whitebox PGD attacks against Prior Network than against baseline models. It is necessary to point out that the evaluation of adversarial attack detection using Prior Networks is limited to only strong $L_{\infty}$ attacks. It is of interest to assess how well Prior Networks are able to detect adaptive C&W $L_2$ attacks [21] and EAD $L_1$ attacks [33].

**Acknowledgments**

This paper reports on research partly supported by Cambridge Assessment, University of Cambridge. This work is also partly funded by a DTA EPSRC award.

## Footnotes

[2]Here $\boldsymbol{\pi} = [\mathrm{P}(y = \omega_1), \cdots, \mathrm{P}(y = \omega_K)]^{\mathrm{T}}$ - the parameters of a categorical distribution.

[3]Alternate choices of distribution, such as a mixture of Dirichlets or the Logistic-Normal, are possible.

[4]Described in appendix B.

[5] An on-going PyTorch re-implementation of this paper, along updated results, is available at https://github.com/KaosEngineer/PriorNetworks

[6]Results are described in appendix E.

[7]The may result in minimum AUROC performance being a little greater than 50 is the success rate is not 100 %, as is the case with MCDP AUROC in figure 5.

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
