[Supplementary Material]

## Appendix A   Further Analysis of reverse KL-divergence Loss

It is interesting to further analyze the properties of the *reverse* KL-divergence loss by decomposing it into the reverse *cross-entropy* and the negative differential entropy:

$$\mathcal{L}^{RKL}(\boldsymbol{\theta};\beta) = \mathbb{E}_{\hat{p}_{tr}(\boldsymbol{x})}\Big[\underbrace{\mathbb{E}_{p(\boldsymbol{\pi}|\boldsymbol{x};\boldsymbol{\theta})}\big[-\ln \text{Dir}(\boldsymbol{\pi}|\bar{\boldsymbol{\beta}})\big]}_{Reverse\ Cross-Entropy} - \underbrace{\mathcal{H}\big[p(\boldsymbol{\pi}|\boldsymbol{x};\boldsymbol{\theta})\big]}_{Differential\ Entropy}\Big] \tag{14}$$

Lets consider the reverse-cross entropy term in more detail (and dropping additive constants):

$$\mathcal{L}^{RCE}(\boldsymbol{\theta};\beta) = \mathbb{E}_{\hat{p}_{tr}(\boldsymbol{x})}\Big[\mathbb{E}_{p(\boldsymbol{\pi}|\boldsymbol{x};\boldsymbol{\theta})}\Big[-\sum_{c=1}^{K}\sum_{k=1}^{K}\hat{P}_{tr}(y=\omega_c|\boldsymbol{x})\big(\beta_k^{(c)}-1\big)\ln\pi_k\Big]\Big]$$

$$= \mathbb{E}_{\hat{p}_{tr}(\boldsymbol{x})}\Big[-\sum_{c=1}^{K}\sum_{k=1}^{K}\hat{P}_{tr}(y=\omega_c|\boldsymbol{x})\big(\beta_k^{(c)}-1\big)\big(\psi(\hat{\alpha}_k)-\psi(\hat{\alpha}_0)\big)\Big] \tag{15}$$

When the target concentration parameters $\boldsymbol{\beta}^{(c)}$ are defined as in equation 7, the form of the reverse cross-entropy will be:

$$\mathcal{L}^{RCE}(\boldsymbol{\theta};\beta) = \mathbb{E}_{\hat{p}_{tr}(\boldsymbol{x})}\Big[-\beta\sum_{c=1}^{K}\hat{P}_{tr}(y=\omega_c|\boldsymbol{x})\big(\psi(\hat{\alpha}_c)-\psi(\hat{\alpha}_0)\big)\Big] \tag{16}$$

This expression for the reverse cross entropy is a scaled version of an upper-bound to the cross entropy between *discrete* distributions, obtained via Jensen's inequality, which was proposed in a parallel work [34] that investigated a model similar to Dirichlet Prior networks:

$$\mathcal{L}^{NLL-UB}(\boldsymbol{\theta}) = \mathbb{E}_{\hat{p}_{tr}(\boldsymbol{x})}\Big[-\sum_{c=1}^{K}\hat{P}_{tr}(y=\omega_c|\boldsymbol{x})\big(\psi(\hat{\alpha}_c)-\psi(\hat{\alpha}_0)\big)\Big] \tag{17}$$

This form of this upper bound loss is identical to standard negative log-likelihood loss, except with digamma functions instead of natural logarithms. This loss can be analyzed further by considering the following asymptotic series approximation to the digamma function:

$$\psi(x) = \ln x - \frac{1}{2x} + \mathcal{O}(x^2) \approx \ln x - \frac{1}{2x} \tag{18}$$

Given this approximation, it is easy to show that this upper-bound loss is equal to the negative log-likelihood plus an extra term which drives the concentration parameter $\hat{\alpha}_c$ to be as large as possible:

$$\mathcal{L}^{NLL-UB}(\boldsymbol{\theta}) = \mathbb{E}_{\hat{p}_{tr}(\boldsymbol{x})}\Big[-\sum_{c=1}^{K}\hat{P}_{tr}(y=\omega_c|\boldsymbol{x})\big(\psi(\hat{\alpha}_c)-\psi(\hat{\alpha}_0)\big)\Big]$$

$$\approx \mathbb{E}_{\hat{p}_{tr}(\boldsymbol{x})}\Big[-\sum_{c=1}^{K}\hat{P}_{tr}(y=\omega_c|\boldsymbol{x})\big(\ln(\hat{\pi}_c)-\frac{1-\hat{\pi}}{2\hat{\alpha}_c}\big)\Big] \tag{19}$$

$$= \mathcal{L}^{NLL}(\boldsymbol{\theta}) + \mathbb{E}_{\hat{p}_{tr}(\boldsymbol{x})}\Big[\sum_{c=1}^{K}\hat{P}_{tr}(y=\omega_c|\boldsymbol{x})\big(\frac{1-\hat{\pi}}{2\hat{\alpha}_c}\big)\Big]$$

Thus, the reverse KL-divergence between Dirichlet distributions, given setting of target concentration parameters via equation 7, yields the following expression:

$$\mathcal{L}^{RKL}(\boldsymbol{\theta};\beta) \approx \beta \cdot \mathcal{L}^{NLL}(\boldsymbol{\theta}) + \mathbb{E}_{\hat{p}_{tr}(\boldsymbol{x})}\Big[\beta\sum_{c=1}^{K}\hat{P}_{tr}(y=\omega_c|\boldsymbol{x})\big(\frac{1-\hat{\pi}}{2\hat{\alpha}_c}\big) - \mathcal{H}\big[p(\boldsymbol{\pi}|\boldsymbol{x};\boldsymbol{\theta})\big]\Big] \tag{20}$$

Clearly, this expression is equal to the standard negative log-likelihood loss for discrete distributions, weighted by $\beta$, plus a term which drives the precision $\hat{\alpha}_0$ of the Dirichlet to be $\beta + K$, where $K$ is the number of classes.

## Appendix B  Synthetic Experiments

The current appendix describes the high data uncertainty artificial dataset used in section 4 of this paper. This dataset is sampled from a distribution $\mathrm{p_{tr}}(\boldsymbol{x}, y)$ which consists of three normally distributed clusters with tied isotropic covariances with equidistant means, where each cluster corresponds to a separate class. The marginal distribution over $\boldsymbol{x}$ is given as a mixture of Gaussian distributions:

$$\mathrm{p_{tr}}(\boldsymbol{x}) = \sum_{c=1}^{3} \mathrm{p_{tr}}(\boldsymbol{x}|y=\omega_c) \cdot \mathrm{P_{tr}}(y=\omega_c) = \frac{1}{3}\sum_{c=1}^{3} \mathcal{N}(\boldsymbol{x}; \boldsymbol{\mu}_c, \sigma^2 \cdot \boldsymbol{I}) \tag{21}$$

The conditional distribution over the classes $y$ can be obtained via Bayes' rule:

$$\mathrm{P_{tr}}(y=\omega_c|\boldsymbol{x}) = \frac{\mathrm{p_{tr}}(\boldsymbol{x}|y=\omega_c) \cdot \mathrm{P_{tr}}(y=\omega_c)}{\sum_{k=1}^{3} \mathrm{p_{tr}}(\boldsymbol{x}|y=\omega_k) \cdot \mathrm{P_{tr}}(y=\omega_k)} = \frac{\mathcal{N}(\boldsymbol{x}; \boldsymbol{\mu}_c, \sigma^2 \cdot \boldsymbol{I})}{\sum_{k=1}^{3} \mathcal{N}(\boldsymbol{x}; \boldsymbol{\mu}_k, \sigma^2 \cdot \boldsymbol{I})} \tag{22}$$

This dataset is depicted for $\sigma = 4$ below. The green points represent the 'out-of-distribution' training data, which is sampled close to the in-domain region. The Prior Networks considered in section 4 are trained on this dataset. Figure 7 depicts the behaviour of the differential entropy of Prior Networks

Figure 6: High Data Uncertainty artificial dataset.

trained on the high data uncertainty artificial dataset using both KL-divergence losses. Unlike the *total uncertainty*, *expected data uncertainty* and mutual information, it is less clear what is the desired behaviour of the differential entropy. Figure 7 shows that both losses yield low differential entropy in-domain and high differential entropy out-of-distribution. However, the reverse KL-divergence seems to capture more of the structure of the dataset, which is especially evident in figure 7b, than the forward KL-divergence. This suggests that the differential entropy of Prior Networks trained via *reverse* KL-divergence is a measures of *total uncertainty*, while the differential entropy of Prior Networks trained using *forward* KL-divergence is a measure of *knowledge uncertainty*. The latter is consistent with results in [16].

(a) Differential Entropy PN-KL          (b) Differential Entropy PN-RKL

Figure 7: Differential Entropy derived from Prior Networks trained with *forward* and *reverse* KL-divergence loss.

# Appendix C  Experimental Setup

The current appendix describes the experimental setup and datasets used for experiments considered in this paper. Table 3 describes the datasets used in terms of their size and numbers of classes.

Table 3: Description of datasets in terms of number of images and classes.

| Dataset | Train | Valid | Test | Classes |
|---|---|---|---|---|
| MNIST | 55000 | 5000 | 10000 | 10 |
| SVHN | 73257 | - | 26032 | 10 |
| CIFAR-10 | 50000 | - | 10000 | 10 |
| LSUN | - | - | 10000 | 10 |
| CIFAR-100 | 50000 | - | 10000 | 100 |
| TinyImagenet | 100000 | 10000 | 10000 | 200 |

All models considered in this paper were implemented in Tensorflow [35] using the VGG-16 [2] architecture for image classification, but with the dimensionality of the fully-connected layer reduced down to 2048 units. DNN models were trained using the negative log-likelihood loss. Prior Networks were trained using both the forward KL-divergence (PN-KL) and reverse KL-divergence (PN-RKL) losses to compare their behaviour on more challenging datasets. Identical target concentration parameters $\beta^{(c)}$ were used for both the forward and reverse KL-divergence losses. All models were trained using the Adam [36] optimizer, with a 1-cycle learning rate policy and dropout regularization. In additional, data augmentation was done when training models on the CIFAR-10, CIFAR-100 and TinyImageNet datasets via random left-right flips, random shifts up to $\pm 4$ pixels and random rotations by up to $\pm$ 15 degrees. The details of the training configurations for all models and each dataset can be found in table 4. 5 models of each type were trained starting from different random seeds. The 5 DNN models were evaluated both individually (DNN) and as an explicit ensemble of models (ENS).

## C.1  Adversarial Attack Generation

An adversarial input $\boldsymbol{x}_{\texttt{adv}}$ will be defined as the output of a constrained optimization process $\mathcal{A}_{\texttt{adv}}$ applied to a *natural* input $\boldsymbol{x}$:

$$\mathcal{A}_{\texttt{adv}}(\boldsymbol{x}, \omega_t) = \arg \min_{\tilde{\boldsymbol{x}} \in \mathcal{R}^D} \left\{ \mathcal{L}\big(y = \omega_t, \tilde{\boldsymbol{x}}, \hat{\boldsymbol{\theta}}\big)\right\} : \; \delta(\boldsymbol{x}, \tilde{\boldsymbol{x}}) < \epsilon \tag{23}$$

The loss $\mathcal{L}$ is typically the negative log-likelihood of a particular target class $y = \omega_t$:

$$\mathcal{L}\big(y = \omega_t, \tilde{\boldsymbol{x}}, \hat{\boldsymbol{\theta}}\big) = -\ln \mathrm{P}(y = \omega_t | \tilde{\boldsymbol{x}}; \hat{\boldsymbol{\theta}}) \tag{24}$$

The distance $\delta(\cdot, \cdot)$ represents a proxy for the *perceptual distance* between the natural sample $\boldsymbol{x}$ and the adversarial sample $\tilde{\boldsymbol{x}}$. In the case of adversarial images $\delta(\cdot, \cdot)$ is typically the $L_1$, $L_2$ or $L_\infty$ norm. The distance $\delta(\cdot, \cdot)$ is constrained to be within the set of allowed perturbations such that the adversarial attack is still *perceived* to be a natural input to a human observer. First-order optimization under a $L_p$ constraint is called Projected Gradient Descent [25], where the solution is projected back onto the $L_p$-norm ball whenever it exceeds the constraint.

There are multiple ways in which the PGD optimization problem 23 can be solved [17, 18, 19, 20, 25]. The simplest way to generate an adversarial example is via the *Fast Gradient Sign Method* or FGSM [18], where the sign of the gradient of the loss with respect to the input is added to the input:

$$\boldsymbol{x}_{adv} = \boldsymbol{x} - \epsilon \cdot \texttt{sign}(\nabla_{\boldsymbol{x}} \mathcal{L}\big(\omega_t, \boldsymbol{x}, \hat{\boldsymbol{\theta}}\big)) \tag{25}$$

Epsilon controls the magnitude of the perturbation under a particular distance $\delta(\boldsymbol{x}, \boldsymbol{x}_{adv})$, the $L_\infty$ norm in this case. A generalization of this approach to other $L_p$ norms, called *Fast Gradient Methods* (FGM), is provided below:

$$\boldsymbol{x}_{adv} = \boldsymbol{x} - \epsilon \cdot \frac{\nabla_{\boldsymbol{x}} \mathcal{L}\big(\omega_t, \boldsymbol{x}, \hat{\boldsymbol{\theta}}\big)}{||\nabla_{\boldsymbol{x}} \mathcal{L}\big(y_t, \boldsymbol{x}, \hat{\boldsymbol{\theta}}\big)||_p} \tag{26}$$

FGM attacks are simple adversarial attacks which are not always successful. A more challenging class of attacks are iterative FGM attacks, such as the Basic Iterative Method (BIM) [19] and Momentum

Table 4: Training Configurations. $\eta_0$ is the initial learning rate, $\gamma$ is the out-of-distribution loss weight and $\beta$ is the concentration of the target class. The batch size for all models was 128. Dropout rate is quoted in terms of probability of *not* dropping out a unit.

| Training Dataset | Model | $\eta_0$ | Epochs | Cycle Length | Dropout | $\gamma$ | $\beta_{in}$ | OOD data |
|---|---|---|---|---|---|---|---|---|
| MNIST | DNN | 1e-3 | 20 | 10 | 0.5 | | | - |
| | PN-KL PN-RKL | | | | | 0.0 | 1e3 | - |
| SVHN | DNN | 1e-3 | 40 | 30 | 0.5 | | | - |
| | PN-KL PN-RKL | 5e-4 5e-6 | | | 0.7 0.7 | 1.0 10.0 | 1e3 | CIFAR-10 |
| CIFAR-10 | DNN DNN-ADV | 1e-3 | 45 | 30 | 0.5 | - | - | - FGSM-ADV |
| | PN-KL PN-RKL | 5e-4 5e-6 | 45 | 30 | 0.7 | 1.0 10.0 | 1e2 | CIFAR-100 |
| | PN | 5e-6 | 45 | 30 | 0.7 | 30.0 | 1e2 | FGSM-ADV |
| CIFAR-100 | DNN DNN-ADV | 1e-3 | 100 | 70 | 0.5 | - | - | - FGSM-ADV |
| | PN-KL PN-RKL | 5e-4 5e-6 | 100 | 70 | 0.7 | 1.0 10.0 | 1e2 | TinyImageNet |
| | PN | 5e-4 | 100 | 70 | 0.7 | 30.0 | 1e2 | FGSM-ADV |
| TinyImageNet | DNN | 1e-3 | 120 | 80 | 0.5 | | | - |
| | PN-KL PN-RKL | 5e-4 5e-6 | | | | 0.0 | 1e2 | - |

Iterative Method (MIM) [20], and others [21, 33]. However, as pointed out by Madry et. al [25], all of these attacks, whether one-step or iterative, are generated using variants of *Projected Gradient Descent* to solve the constrained optimization problem in equation 23. Madry [25] argues that all attacks generated using various forms of PGD share similar properties, even if certain attacks use more sophisticated forms of PGD than others.

In this work MIM $L_\infty$ attacks, which are considered to be strong $L_\infty$ attacks, are used to attack all models considered in section 6. However, standard targeted attacks which minimize the negative log-likelihood of a target class are not *adaptive* to the detection scheme. Thus, in this work *adaptive* targeted attacks are generated by minimizing the losses proposed in section 6, in equation 13.

The optimization problem in equation 23 contains a *hard constraint*, which essentially projects the solutions of gradient descent optimization to the allowed $L_p$-norm ball whenever $\delta(\cdot, \cdot)$ is larger than the constraint. This may be both disruptive to iterative momentum-based optimization methods. An alternative *soft-constraint* formulation of the optimization problem is to simultaneously minimize the loss as well as the perturbation $\delta(\cdot, \cdot)$ directly:

$$\mathcal{A}_{\text{adv}}(\boldsymbol{x}, t) = \arg \min_{\tilde{\boldsymbol{x}} \in \mathcal{R}^K} \left\{ \mathcal{L}(\omega_t, \tilde{\boldsymbol{x}}, \hat{\boldsymbol{\theta}}) + c \cdot \delta(\boldsymbol{x}, \tilde{\boldsymbol{x}}) \right\} \quad (27)$$

In this formulation $c$ is a hyper-parameter which trades of minimization of the loss $\mathcal{L}(\omega_t, \tilde{\boldsymbol{x}}, \hat{\boldsymbol{\theta}})$ and the perturbation $\delta(\cdot, \cdot)$. Approaches which minimize this expression are the Carlini and Wagner $L_2$ (C&W) attack [21] and the "Elastic-net Attacks to DNNs" (EAD) attack [33]. While the optimization expression is different, these methods are also a form of PGD and therefore are expected to have similar properties as other PGD-based attacks [25]. The C&W and EAD are considered to be particularly strong $L_2$ and $L_1$ attacks, and Prior Networks need to be assessed on their ability to be robust to and detect them. However, adaptation of these attacks to Prior Networks is non-trivial and left to future work.

## C.2 Adversarial Training of DNNs and Prior Networks

Prior Networks and DNNs considered in section 6 are trained on a combination of natural and adversarially perturbed data, which is known as adversarial training. DNNs are trained on $L_\infty$ targeted FGSM attacks which are generated dynamically during training from the current training minibatch. The target class $\omega_t$ is selected from a uniform categorical distribution, but such that it is **not** the true class of the image. The magnitude of perturbation $\epsilon$ is randomly sampled for each image in the minibatch from a truncated normal distribution, which only yields positive values, with a standard deviation of 30 pixels:

$$\epsilon \sim \mathcal{N}_{pos}(0, \frac{30}{128}) \tag{28}$$

The perturbation strength is sampled such that the model learns to be robust to adversarial attacks across a range of perturbations. The DNN is then trained via maximum likelihood on both the natural and adversarially perturbed version of the minibatch.

Adversarial training of the Prior Network is a little more involved. During training, an adversarially perturbed version of the minibatch is generated using the targeted FGSM method. However, the loss is not the negative log-likelihood of a target class, but the reverse KL-divergence (eqn. 11) between the model and a targeted Dirichlet which is focused on a target class which is chosen from a uniform categorical distribution (but not the true class of the image). For this loss the target concentration is the same as for natural data ($\beta_{in} = 1e2$). The Prior Network is then jointly trained on the natural and adversarially perturbed version of the minibatch using the following loss:

$$\mathcal{L}(\boldsymbol{\theta}, \mathcal{D}) = \mathcal{L}_{in}^{RKL}(\boldsymbol{\theta}, \mathcal{D}_{train}; \beta_{in}) + \gamma \cdot \mathcal{L}_{adv}^{RKL}(\boldsymbol{\theta}, \mathcal{D}_{adv}; \beta_{adv}) \tag{29}$$

Here, the concentration of the target class for natural data is $\beta_{in} = 1e2$ and for adversarially perturbed data $\beta_{adv} = 1$, where the concentration parameters are set via 7. Setting $\beta_{adv} = 1$ results in a very wide Dirichlet distribution whose mode and mean are closest to the target class. This ensures that the prediction yields the correct class and that all measure of uncertainty, such as entropy of the predictive posterior or the mutual information, are high. Note, that due to the nature of the reverse KL-divergence loss, adversarial inputs which have a very small perturbation $\epsilon$ and lie close to their natural counterparts will naturally have a target concentration which is an interpolation between the concentration for natural data and for adversarial data. The degree of interpolation is determined by the OOD loss weight $\gamma$, as discussed in section 3.

It is necessary to point out that FGSM attack are used because they are computationally cheap to compute during training. However, iterative adversarial attacks can also be considered during training, although this will make training much slower.

## Appendix D   Jointly Assessing Adversarial Attack Robustness and Detection

In order to investigate detection of adversarial attacks, it is necessary to discuss how to assess the effectiveness of an adversarial attack in the scenario where detection of the attack is possible. Previous work on detection of adversarial examples [37, 38, 39, 14, 15] assesses the performance of detection methods separately from whether an adversarial attack was successful, and use the standard measures of adversarial success and detection performance. However, in a real deployment scenario, an attack can only be considered successful if it *both* affects the predictions *and* evades detection. Here, we develop a measure of performance to assess this.

For the purposes of this discussion the adversarial generation process $\mathcal{A}_{\mathrm{adv}}$ will be defined to either yield a successful adversarial attack $\boldsymbol{x}_{\mathrm{adv}}$ or an empty set $\emptyset$. In a standard scenario, where there is no detection, the efficacy of an adversarial attack on a model[8] can be summarized via the *success rate* $\mathcal{S}$ of the attack:

$$\mathcal{S} = \frac{1}{N} \sum_{i=1}^{N} \mathcal{I}(\mathcal{A}_{\mathrm{adv}}(\boldsymbol{x}_i, \omega_t)), \quad \mathcal{I}(\boldsymbol{x}) = \begin{cases} 1, \ \boldsymbol{x} \neq \emptyset \\ 0, \ \boldsymbol{x} = \emptyset \end{cases} \tag{30}$$

Typically $\mathcal{S}$ is plotted against the total maximum perturbation $\epsilon$ from the original image, measured as either the $L_1$, $L_2$ or $L_\infty$ distance from the original image.

Consider using a threshold-based detection scheme where a sample is labelled 'positive' if some measure of uncertainty $\mathcal{H}(\boldsymbol{x})$, such as entropy or mutual information, is less than a threshold $T$ and 'negative' if it is higher than a threshold:

$$\mathcal{I}_T(\boldsymbol{x}) = \begin{cases} 1, \ T > \mathcal{H}(\boldsymbol{x}) \\ 0, \ T \leq \mathcal{H}(\boldsymbol{x}) \end{cases} \tag{31}$$

The performance of such a scheme can be evaluated at every threshold value using the *true positive rate* $t_p(T)$ and the *false positive rate* $f_p(T)$:

$$t_p(T) = \frac{1}{N} \sum_{i=1}^{N} \mathcal{I}_T(\boldsymbol{x}_i), \quad f_p(T) = \frac{1}{N} \sum_{i=1}^{N} \mathcal{I}_T(\mathcal{A}_{\mathrm{adv}}(\boldsymbol{x}_i, \omega_t)) \tag{32}$$

The whole range of such trade offs can be visualized using a Receiver-Operating-Characteristic (ROC) and the quality of the trade-off can be summarized using area under the ROC curve. However, a standard ROC curve does account for situations where the process $\mathcal{A}_{\mathrm{adv}}(\cdot)$ fails to produce a successful attack. In fact, if an adversarial attack is made against a system which has a detection scheme, it can only be considered successful if it *both* affects the predictions *and* evades detection. This condition can be summarized in the following indicator function:

$$\hat{\mathcal{I}}_T(\boldsymbol{x}) = \begin{cases} 1, \ T > \mathcal{H}(\boldsymbol{x}) \\ 0, \ T \leq \mathcal{H}(\boldsymbol{x}) \\ 0, \ \boldsymbol{x} = \emptyset \end{cases} \tag{33}$$

Given this indicator function, a new false positive rate $\hat{f}_P(T)$ can be defined as:

$$\hat{f}_p(T) = \frac{1}{N} \sum_{i=1}^{N} \hat{\mathcal{I}}_T(\mathcal{A}_{\mathrm{adv}}(\boldsymbol{x}_i, \omega_t)) \tag{34}$$

This false positive rate can now be seen as a new *Joint Success Rate* which measures how many attacks were both successfully generated and evaded detection, given the threshold of the detection scheme. The *Joint Success Rate* can be plotted against the standard true positive rate on an ROC curve to visualize the possible trade-offs. One possible operating point is where the false positive rate is equal to the false negative rate, also known as the *Equal Error-Rate* point:

$$\hat{f}_P(T_{\mathrm{EER}}) = 1 - t_P(T_{\mathrm{EER}}) \tag{35}$$

Throughout this work the EER false positive rate will be quoted as the *Joint Success Rate*.

# Appendix E  Additional Adversarial Attack Detection Experiments

In this appendix additional experiments on adversarial attack detection are presented. In figure 8 adaptive whitebox adversarial attacks generated by iteratively minimizing KL divergence between the original and target (permuted) categorical distributions $\mathcal{L}_{PMF}^{KL}$ are compared to attacks generated by minimzing the KL-divergence between the predicted and permuted Dirichlet distributions $\mathcal{L}_{DIR}^{KL}$. Performance is assessed only against Prior Network models. The results show that KL PMF attacks are more successful at switching the prediction to the desired class and at evading detection. The could be due to the fact that Dirichlet distributions which are sharp at different corners have limited common support, making the optimization of the KL-divergence between them more difficult than the KL-divergence between categorical distributions.

(a) C10 Success Rate    (b) C10 ROC AUC    (c) C10 Joint Success Rate

Figure 8: Comparison of performance of whitebox adaptive PGD MIM $L_\infty$ attacks which minimize the KL-divergence between PMFs (KL PMF) and Dirichlet distributions (KL DIR) on CIFAR-10.

Results in figure 9 show that $L_2$ PGD Momentum Iterative attacks which minimize the $\mathcal{L}_{PMF}^{KL}$ loss are marginally more successful than the $L_\infty$ version of these attacks. However, it is necessary to consider appropriate adaptation of the C&W $L_2$ attacks to the loss functions considered in this work for a more aggressive set of $L_2$ attacks.

(a) C10 Success Rate    (b) C10 ROC AUC    (c) C10 Joint Success Rate

(d) C100 Success Rate    (e) C100 ROC AUC    (f) C100 Joint Success Rate

Figure 9: Comparison of performance of whitebox adaptive $L_\infty$ and $L_2$ PGD MIM attacks against Prior Networks trained on CIFAR-10 (C10) and CIFAR-100 (C100) datasets.

## Footnotes

[8]Given an evaluation dataset $\mathcal{D}_{\mathrm{test}} = \{\boldsymbol{x}_i, y_i\}_{i=1}^{N}$