[Reviews · NeurIPS 2019]

Reviewer 1



This paper addresses an interesting problem and proposes a solution that is shown to have empirical advantages. However, the text of the paper has been poorly executed. The notation has not been clearly defined or explained. In line 75, $x*$, $\hat{\theta}$ are not defined. These variables have been consistently used in the rest of the paper. I had to read [18] to understand this notation. In lines 74-80 the definition of a prior network is unclear. In lines 196-199 the intuitive explanation for why prior networks are more robust to adversarial attacks is also unclear. This diminishes the quality of this paper as a standalone piece of work. The main contribution of this work is the improved training criterion. In previous work, prior networks were trained under the forward KL divergence while this paper proposes to use the reverse KL divergence instead. This implies empirical benefits in training. It is also shown empirically that these networks have better out of distribution detection performance and in some cases are shown to be more robust to adversarial attacks. However, in complex datasets like CIFAR-100 the improvement shown is only modest, so it would be nice to see the performance of these networks on more datasets (like ImageNet). ------------------------------------------------------------------------------------------------------------------------------------------------ In light of the author response I tend to keep my overall score (6).

Reviewer 2



As described above, the paper makes an important observation regarding the implications of training with a forwards vs. reverse KL loss in prior networks. The analysis is compelling, and nicely explains prior observations of the difficulty for training prior networks on datasets with many classes. It’s nice to see that with the reverse KL loss, this is now possible. My main concerns are with the empirical evaluation, and whether the paper provides convincing evidence that prior networks trained with reverse KL are SOTA for the two discussed tasks. For OOD detection: All the experiments currently are run on re-implementations. It’s a standard enough task - why not compare to previously reported numbers? Relatively minor: The network performances are noticeably sub-par for 2019. Even setting aside tricks, and data augmentation, 8.0% for a DNN baseline on CIFAR-10 is twice of 4.0% for a wide resnet, no tricks. For adversarial example detection: Again, it would be nice to compare to previously reported numbers, rather than comparing only to re-implementations. Relatedly, an L_inf bound of 30 pixels is a strange choice. eps=8/255 is really standard on CIFAR-10, and seems to be the natural choice to allow comparison with prior work. Relatively minor: It would be nice in addition to reporting AUROC, to report joint success rate for a fixed recall (say, 80%, 95%, and 99%), to allow comparison to adversarial accuracy rates without detection. Against strong whitebox attacks, the joint success rate is extremely high (appears >90%). Additionally, MC-Dropout appears to outperform the proposed prior networks approach. In general, demonstrating that finding undetected adversarial examples “takes a greater amount of computational effort” is not considered a significant result, as it is also possible purely through masking gradients without improving the robustness of the model. It’s nice to see that the paper implements an adaptive attack for the model under consideration, and reports metrics against the strongest attack they find. Minor comments: For Figure 2, it would be nice to include the data distribution for convenience, so that readers don’t need to refer to ref. 16. Why not use a stronger attack than FGSM for adversarial training? In the second line of Eq. 10, I believe there’s a missing normalization constant. This doesn’t affect the correctness of the argument. I couldn't find the code referenced in the paper or supplement, though the reproducibility checklist said it was included. My apologies if I missed it. I think it's great to share this though - especially for the adversarial robustness community, it's really helpful in ensuring proposed defenses are truly robust. ________________ Update Thanks to the authors for providing additional insights. The original paper and responses still do not provide convincing empirical evidence that prior networks with the reverse KL loss are improvements for existing approaches for adversarial robustness. The OOD evaluation also still has weaknesses. I'm glad that you're making additional preparations to run other experiments, but it's somewhat disappointing that it wasn't possible to re-run experiments under standard settings (e.g. on CIFAR-10, no work uses eps=30, and eps=8 is standard across the field), which makes it very difficult to compare this to prior work. Similarly, attack success at fixed recall would be easier to compare to e.g. Madry adversarial training / TRADES / other adversarial robustness work, though this is less significant. Increasing required number of attack iterations is typically not considered a significant result within the adversarial examples community, even without additional training cost. See e.g. Athalye et al, ICML 2018, or Carlini et al 2019 on adversarial evaluation. Joint success rate: My mistake, thank you for the clarification. I was looking at the "success rate" column where MC-Dropout outperforms prior networks (for large # of iterations, the regime of interest). RKL-PN indeed outperforms significantly under JSR. This point does make me more positive, though not sufficiently to raise my score. For OOD detection: It's fine not to compare to bespoke post-processing techniques which use domain-specific knowledge, but that doesn't seem like a good justification to not compare to previously reported numbers at all. (I don't understand why the rebuttal includes classification accuracy numbers, rather than OOD detection numbers. The question of interest is whether the technique can outperform existing OOD detection, when using standard architectures.) Again, I want to emphasize that I think this is good work, and could be impactful within the uncertainty/OOD detection and adversarial robustness communities. I don't think the current evaluation in the paper is sufficient though, and the rebuttal does not provide any experiments to address this, so I believe the paper would be better served with a stronger empirical evaluation.

Reviewer 3



The authors present a novel algorithm with theoretical analysis and empirical results. We have a few comments and suggestions for the work: The comparison of forward vs reverse KL divergence as the objective criteria resembles the choice of mode vs. mean seeking form of the objective in variational inference, respectively (in this case applied with a Dirichlet distribution). We recommend that the authors refer and make connections to this similar literature. It would be great if the authors could expand upon the distinction of in-domain and out-of-domain training data in lines 105-106. How are these datasets created and is the purpose of separating the data to improve generalization. How is the optimization performed in practice? In the algorithm, the authors propose to set the in-domain \beta parameters to large value of 1e2 and the out-of-domain parameters to small values of 0. How sensitive are the results to these specific choices? The authors also note that the losses were equally weighted using the forward KL divergence and had a large relative weighting \gamma when using the reverse loss. What criteria was used to optimally choose the \gamma parameter? Lastly, we had a few minor suggestions for the text: using the conventional indicator variable instead of \mathcal{I} may be more clear, and defining all notation (i.e., \pi) in the main text may improve readability.

[Author Response · NeurIPS 2019]

Thank you for yours detailed reviews. We would first like to address the **general concerns**. Firstly, we agree that the clarity and writing of the paper needs improvement, both in terms of notation and explanation of various terms, and especially in the section on adversarial attack detection. We have taken your comments on board and are currently addressing these issues in order to make the story of the paper far clearer and easier to follow. Secondly, we agree that the experimental evaluation needs improvement. This was mostly due to having legacy tensorflow code that was not suitable for extension. The last few months have been spent writing a cleaner PyTorch implementation which allows for the easier introduction of new architecures and integration with adversarial toolkits (FoolBox/Cleverhans) and allows for a head-to-head comparison of various methods. We are currently reproducing (classification accuracy results presented in table below) the OOD detection experiments using the WideResnet 28x10 architecture, where we are able to match SOTA classification performance on CIFAR-10, CIFAR-100 and TinyImageNet. Additionally, we will add experiments on RKL PNs trained on TinyImageNet in-domain and a 400-class subset of the other 800 imageNet classes, processed like TinyImageNet, as OOD training data. We will evaluate OOD detection on the remaining, heldout subset of 400 ImageNet classes. Third, we will update adversarial attack detection numbers to be on the CIFAR-10, CIFAR-100 and TinyImageNet datasets using the new architecture. Furthermore, having analyzed the Carlini&Wagner L2 attack, we believe that the current adaptive adversarial attack loss functions may be sufficient. An alternative adaptive attack loss function we will also consider is L1 loss between the predicted and permuted logits, which has the same fixed point as KL-divergence minimization, but potentially nicer gradients. This should yield an evaluation against a stronger adaptive whitebox adversary. However, we are unable to demonstrate updated Adversarial Attack Detection numbers in this rebuttal as we haven't integrated with Foolbox yet.

| Model | CIFAR-10 | CIFAR-100 | TinyImageNet |
|---|---|---|---|
| VGG (Error %) | 8.0 $\pm 0.4$ | 30.4 $\pm 0.6$ | 41.7 $\pm 0.4$ |
| WRN (Error %) | 3.9 $\pm 0.1$ | 19.3 $\pm 0.5$ | 32.3 $\pm$ NA |

**Reviewer 2** The OOD results quoted in the papers use a range of classification network architectures. It is thus not appropriate to directly compare numbers. We are planning to do clean comparisons of the method using consistent architecture. Secondly, as presented in this paper, Prior Networks are *not* SOTA (but close) for OOD detection, as the current SOTA (Lee18) results make use of a set of *bespoke* post-processing techniques (such as ODIN) aimed at maximizing OOD detection performance. The same approaches can be applied on top of Prior Networks. The aim of this paper is to present a *general method* (training criterion) and analyze its properties in two different scenarios. Crucially, we show that adversarial training of Prior Networks using the proposed criterion gives a significant improvement over standard adversarial training *at not additional cost*, making it a drop-in replacement. Other techniques can be stacked on top of this. Third, a large perturbation size was selected in order to give more freedom to the adversarial attack to succeed against our detection scheme. In additional results to be completed we will conduct an analysis of the perturbation size on the success of adaptive whitebox attacks. Fourth, perhaps we have misunderstand your comment, but the joint success rate represents the success of *the attack*, rather than *the defense*. Specifically, the JSR represents the success of the attack at *both* successfully attaining the target class *and* avoiding detection. This will be made clearer in the text. Thus, it is unsurprising that an adaptive whitebox adversarial attack will have a very high success rate. Fifth, regarding the dropout attack. We need to make it clear in the text that attacks against dropout are completely undetectable, but have a slightly lower success rate since we generate the attack against the *mean network*, rather than against each of the 10 samples. The added stochasticity makes the attack less successful. This was difficult to address in the legacy code, but should be easier to do in the new implementation. Sixth, while we agree that the computational expense of an adversarial attack can be increased by gradient masking, we think that it is a significant result that we can do the same by essentially using an improved form of adversarial training at no additional expense. We stress that we analyse a general method. Other, more task-specific techniques can always be stacked on top of this. Finally, we use FGSM attack in training because we dynamically generate the attack on each minibatch during training. Using iterative attacks is possible, but expensive, as training would slow down.

**Reviewer 3** Firstly, the Reverse KL is exactly equal to the variational criterion, where the reconstruction loss is weighted by $\beta$ and where the KL-regularization loss is minimization the KL-divergence to a flat Dirichlet Prior. We will make this connection clearer in the paper. Secondly, this choice of OOD training data is quite standard in OOD detection experiments. The main requirement is that it is more diverse than the in-domain data. Third, the effect of varying $\beta$ is not strong. The main goal is to make sure that the distribution of $\hat{\alpha}_0$ in-domain and OOD are clearly separable. It is necessary to set $\beta$ to 0 for OOD training data (for OOD detection), as we have no a-priori knowledge about what the target class. However $\beta$ can be set to 1 for adversarial data, as we do know what the target class should be. This allows the model to learn to *both* predict the correct target class *and* high uncertainty for adversarially perturbed training data. Finally, setting $\gamma = 10.0$ for the RKL loss was chosen based on results of the toy-data experiments. When training models on the other datasets we found that this choice of gamma was consistently better than setting gamma to 1.0 .

[Meta-Review · NeurIPS 2019]

Detecting inputs that are outside the distribution of training examples, including adversarial inputs, is an important problem; reviewers and the area chair agree that this paper makes a useful algorithmic contribution towards solving this problem. The argument that reverse KL is conceptually correct, while forward KL as used previously is conceptually wrong, is significant. Training with reverse KL is a simple and compelling idea that practitioners can try easily. For these reasons the paper is being accepted so that the community can benefit from it quickly, despite the fact that reviewers have identified ways in which the writing of the paper, and the empirical evaluation, need improvement. The authors are encouraged to improve the final version.